# Theoretical and Experimental Studies of Micro-Surface Crack Detections Based on BOTDA

**DOI:** 10.3390/s22093529

**Published:** 2022-05-06

**Authors:** Baolong Yuan, Yu Ying, Maurizio Morgese, Farhad Ansari

**Affiliations:** 1College of Information and Control Engineering, Shenyang Jianzhu University, Shenyang 110168, China; kanfenglong@sjzu.edu.cn; 2Department of Civil and Materials Engineering, University of Illinois at Chicago, 842 W Taylor St., Chicago, IL 60607, USA; mmorge3@uic.edu (M.M.); fansari@uic.edu (F.A.)

**Keywords:** distributed optical fiber sensor, micro-cracks, Brillouin frequency

## Abstract

Micro-surface crack detection is important for the health monitoring of civil structures. The present literature review shows that micro-surface cracks can be detected by the Brillouin scattering process in optical fibers. However, the existing reports focus on experiment research. The comparison between theory and experiment for Brillouin-scattering-based optical sensors is rarely reported. In this paper, a distributed optical fiber sensor for monitoring micro-surface cracks is presented and demonstrated. In the simulation, by using finite element methods, an assemblage of a three-dimensional beam model for Brillouin optical time domain analysis (BOTDA) was built. The change in Brillouin frequency (distributed strain) as a function of different cracks was numerically investigated. Simulation results indicate that the amplitudes of the Brillouin peak increase from 27 με to 140 με when the crack opening displacement (COD) is enlarged from 0.002 mm to 0.009 mm. The experiment program was designed to evaluate the cracks in a beam with the length of 15 m. Experimental results indicate that it is possible to detect the COD in the length of 0.002~0.009 mm, which is consistent with the simulation data. The limitations of the proposed sensing method are discussed, and the future research direction is prospected.

## 1. Introduction

Optical fiber sensors have been widely used in structural health monitoring for judging the growth process of cracks in buildings, bridges and other forms of civil infrastructure [1,2,3,4,5,6]. Compared with a traditional electrical sensor, the optical fiber sensor has a lot of advantages, such as anti-corrosion resistance, anti-electromagnetic interference, and high sensitivity. Based on these advantages, optical fiber sensors have been attaining a lot of interest from researchers and engineers in the past few years [7,8,9,10,11].

At present, there are three main optical fiber sensors for structural health monitoring: the local optical fiber sensor, the quasi-distributed optical fiber sensor, and the distributed optical fiber sensor [12,13,14,15,16]. In the aspect of the local optical fiber sensor, Chen H presented an extrinsic Fabry–Perot interferometric (EFPI) with dual cavities to measure the pressure (0.1–3 MPa) and temperature (20–350 °C) [17]. Ghildiyal, S designed an FPI pressure sensor with a copper–beryllium alloy (CBA) diaphragm [18]. A sensitivity of more than 1 µm/bar was obtained. In the aspect of the quasi-distributed optical fiber sensor, Liu, M prepared a fiber Bragg grating (FBG) pressure sensor and encapsulated it in polymer. A sensitivity of 51.296 pm/MPa was experimentally demonstrated in the range of 0~15.5 MPa [19]. Guo, G proposed a multiplexed FBG to detect the strain at discrete locations in the fabric. The average lower error rate of 5.9% can be achieved [20]. In the aspect of the distributed optical fiber sensor, Scarella, A presented an optical fiber sensor based on stimulated Brillouin scattering to measure the strain in the bridge model, where the strain increased monotonously with the crack length [21]. Oskoui, E detected five locations of cracks by using Brillouin scattering [22]. Generally speaking, local optical fiber sensors have been limited to a short distance. Quasi-distributed sensors rely on the prior knowledge of crack location. Distributed optical fiber sensors have the advantages of low cost, long distance and, independence from the prior knowledge of crack position. Among them, the use of the distributed optical fiber sensor is a common method for detecting strain caused by crack-growth damages. The method is based on Brillouin scattering, which provides the local strain information in each spatial resolution along the optical fiber. The interaction between light and phones results in Brillouin frequency, which is linearly dependent on strain. This strain will increase as the crack grows. By measuring the Brillouin frequency shift in the optical fiber, the strain data can be acquired, and the crack growth can be identified. In recent years, some significant progress has been made with the development of measurement technology of Brillouin frequency. Yang, D studied a plastic optical fiber sensing technology combining a signal processing method to detect cracks [23]. The results show that the remarkable resemblance in terms of cracks can be identified. Cheng, L designed a high-precision fiber macro-bending loss crack sensor [24]. The macro-bend loss is linearly related to the crack length. Song, Q analyzed a deep learning method and used it for micro-crack detection [25]. The result shows that the crack width of nearly 23 μm can be accurately monitored. Bassil, A presented a multilayer optical fiber to monitor the opening crack in concrete structures [26], with which a relative error as low as 2% can be obtained. This research contributes some novel methods to structural health monitoring. However, most reports are only based on experimental analysis. Less reports focus on the comparison of theory and experiment. Even in the existing theory of crack detection, only the strain exponential model is discussed [26], and it cannot accurately describe the relationship between strain and crack growth. The finite element method of optical fiber can improve the accuracy of analysis, but the analyses of crack detection by use of three-dimensional optical fiber models are fewer. In addition, the tolerance of optical fiber is worth studying.

In this paper, a three-dimensional beam finite element analysis model is proposed, and the strain as a function of crack opening displacement (COD) is analyzed. An experiment program with two CODs, including Brillouin optical time domain analysis (BOTDA) for distributed detection of strain and cracks, is established to evaluate the feasibility. The COD detection range is studied by analyzing the non-linearity of optical fiber.

## 2. Model Analysis

Accordingly, the present study proposed an optical fiber sensor adhered on a crack substrate (steel beam). Figure 1 shows the distribution optical fiber sensor with two cracks. The theoretical model contains two cracks (COD = 2*δ*_1_ and COD = 2*δ*_2_) along the optical fiber at *Z* = *Z*_1_ = 4.4 m and *Z* = *Z*_2_ = 10.6 m. The fiber core diameter is 9 μm, the fiber cladding is 125 μm, and the fiber coating is 250 μm.

The system based on Brillouin scattering is achieved by scanning the frequency shift in the optical fiber. External strain applied to the optical fiber can lead to Brillouin frequency shift, and the Brillouin frequency shift *Ω_B_*(*Z*) as a function of strain is expressed by [27]
*Ω_B_*(*Z*) = *Cε*Δ*ε*(*Z*) + *C_T_*Δ*T*(*Z*)(1)
where Δ*ε*(*Z*) and Δ*T*(*Z*) refer to the change in strain and temperature, and *Cε* = 0.05 MHz/με and *C_T_* = 1 MHz/°C are Brillouin factors for strain and temperature, respectively.

The strain with the crack in the distributed optical fiber is expressed by [28]
(2)$$\varepsilon (z)=\{\begin{array}{c} \delta_{1}\beta \exp[\beta (Z-Z_{1})]+\delta_{2}\beta \exp[\beta (Z-Z_{2})]\begin{array}{cc} & Z<Z_{1}<Z_{2} \end{array}\\ \delta_{1}\beta \exp[-\beta (Z-Z_{1})]+\delta_{2}\beta \exp[\beta (Z-Z_{2})]\begin{array}{cc} & Z_{1}<Z<Z_{2} \end{array}\\ \delta_{1}\beta \exp[-\beta (Z-Z_{1})]+\delta_{2}\beta \exp[-\beta (Z-Z_{2})]\begin{array}{cc} & Z_{1}<Z_{2}<Z \end{array} \end{array}$$
where *β* is the shear lag factor, which was 25/m consulted in the reference [29]. Figure 2 shows the finite element model in which the SMF28 optical fiber was used. The region may have to be divided into 23,546 mesh elements. The steel beam is a rigid body, so it was not used in the model. The cracks were applied to the bottom surface of the glue layer. Since the crack is axisymmetric, the middle section was built in the COMSOL software. The size parameter is *H*_g_ = 6 mm, *W*_g_ = 10 mm, and *L*_g_ = 106 mm. The mechanical properties of the materials can be consulted in the reference [28].

In order to evaluate the feasibility of the presented BOTDA sensor in the long beam with cracks, an experimental program was designed. Figure 3 shows a test bed, which was fabricated to apply the bending test in a 15 m length of beam. The beam was comprised of three sections at the length of 4.4 m, 6.2 m, and 4.4 m, and had two spliced points. The three sections of the beam were connected by bolts and plates. The beam was supported at two points with the span, and the force was applied at the ends. The damage could be fabricated in each splice joint through tightening or loosening the bolts at the plates. In order to monitor the crack opening displacements caused by loosening the bolts, an arch FBG displacement sensor was used. The detailed fabrication method can be consulted in the reference [30]. A single-mode optical fiber (SMF-28) was used as the sensing optical fiber over the beam. In order to maintain stability, the optical fiber was adhered by glue (epoxy resin) onto the top surface of the beam. An available BOTDA measuring system (Neubrex NBX-6055) was used for measuring the strain of the optical fiber. The BOTDA measuring system employs two light sources, including a pump light and a probe light. They transmit in two opposite directions via the separated optical fiber. The sampling interval and spatial resolution were set as SI = 5 cm and SR = 10 cm. The dynamic measurement was conducted at 26 Hz speed (26 measurements/s).

Table 1 shows the material parameters of the single-mode optical fiber (single-mode transmission) which is considered a distributed sensor.

## 3. Results and Discussion

The theoretical and experimental results were studied and compared. For the simulation model, the strain distribution in the optical fiber with two cracks was studied. The distributed optical fiber sensor were assumed to undergo 49 N, 98 N, 196 N, and 392 N, and the crack opening displacements (CODs) were set as 0.002 mm, 0.006 mm, 0.009 mm, and 0.011 mm. Figure 4a shows the Brillouin frequency shift by numerical integration. It can be seen that when the applied force was increased from 49 N to 392 N, the strain in the middle section increased from 10.3 με to 60 με. Two distinct peaks appeared in the Brillouin frequency spectrum due to the two cracks. The amplitudes of the Brillouin peaks increased from 27 με to 140 με when the CODs at the crack’s location rose from 0.002 mm to 0.011 mm. In order to verify the theoretical analysis, distributions of strain were measured for the four CODs ranging from 0.002 mm to 0.011 mm. Figure 4b shows the distributions of strain along the length of the optical fiber from the experiments after filtering out the noise.

It seems that the theoretical results from Figure 4a and the experimental results from Figure 4b are in line with the location of the crack. However, the difference between amplitudes of strain in no-crack and crack regions can be observed when the simulation results are compared with the experimental results. Compared with the actual measurement value, the theoretical simulations underestimate the amplitudes of strain. Figure 5a,b show the difference, respectively. One of the reasons for this difference can be attributed to the crack displacement of the optical fiber. When the crack occurs, the displacement was detected by FBG. However, the actual displacement of the optical fiber was larger than that detected by FBG. Because the optical fiber was not completely attached to the beam surface, there was still a gap above the beam surface. Therefore, the strain was larger than that in the experiment due to the longer crack opening displacement. The other reason is attributed to the fluctuation of the Brillouin scattering gain spectrum. Sometimes, the inconsistencies associated with the probe light and pump light may occur due to the noise level in many measurements.

In order to ensure that the optical fiber can be re-used, the experiment was conducted for a second time. Figure 6 shows the comparison of the peaks in the crack strain region for the first time and second time. It was seen that the strain decreases at first and then increases along the crack region. This is attributed to the non-linearity of Young’s modulus of the optical fiber [31]. The strain response of optical fiber under continuous loading can be divided into two phases. In the first phase, the optical fiber can exhibit full strain recovery. In the second phase, residual strain occurs in the optical fiber and increases when the strain exceeds a certain threshold value. Figure 7 shows the relation between stress σ and strain ε for distributed optical fiber adhered to the beam surface. When the optical fiber performs in the red region (OA line), the stress and strain exhibits a linear trend. When the strain is larger than the yield stress, the optical fiber performs under a non-linear condition (AB line). In the two ranges of OA and AB, the optical fiber can return to its original memory shape once the stress is released. When the stress exerted upon the optical fiber reaches up to the value of spot B, the optical fiber is physically damaged and cannot return to its original shape, even in the no-stress condition. There will be plastic strain (OC line). In the experiment, due to the excess deformation of the optical fiber, the optical fiber was stretched again, and the stress in the central position decreased. Therefore, the crack below 0.009 mm can be used for the distributed optical fiber through testing the optical fiber characteristic.

## 4. Conclusions

This paper reports a method for COD determination based on calculating the Brillouin frequency peak. The work involved a theoretical simulation as well as an experimental investigation. In the theoretical perspective, FEM was used for determining the distributed strain along the optical fiber. The simulation results indicate that the Brillouin peak frequency increases from 27 με to 140 με with an increase in COD from 0.002 mm to 0.009 mm. In the experimental perspective, a 15 m length of steel beam was designed to realize the distributed strain measurement and simulate crack detection. The experimental results are very consistent with the simulation data. In addition, the nonlinear tolerance of the proposed optical fiber was analyzed.

In summary, the proposed Brillouin distributed optical fiber sensor is applicable for detection and monitoring of crack damage in steel beams, with the advantages of long-distance monitoring, low cost, and simple structure. However, this work is in its initial stage, and studies some basic problems in theoretical and experimental aspects. In the future, some research directions require more attention. For example, firstly, optimizing characteristics and practical application. Secondly, the stress–strain non-linearity of the optical fiber core needs to be studied deeply. Thirdly, the influence of the protective coating and glue on strain transferring to the core is a factor worth considering. Fourthly, the range extension of crack detection needs to be further investigated. Finally, it is necessary to consider the error of the optical fiber due to environmental disturbances.

## Figures and Tables

**Figure 1 sensors-22-03529-f001:**
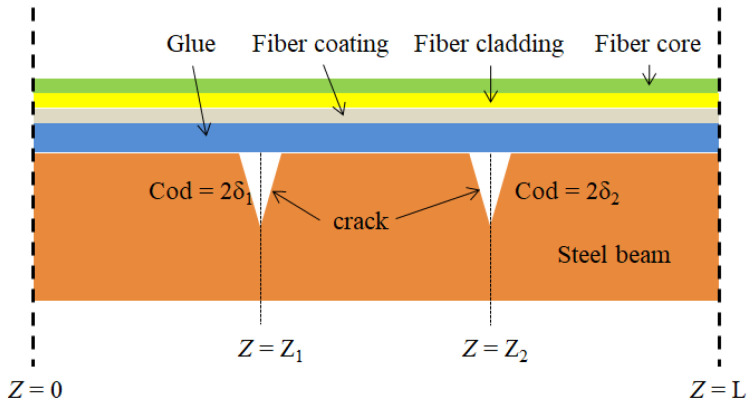
Distribution optical fiber sensor with two cracks.

**Figure 2 sensors-22-03529-f002:**
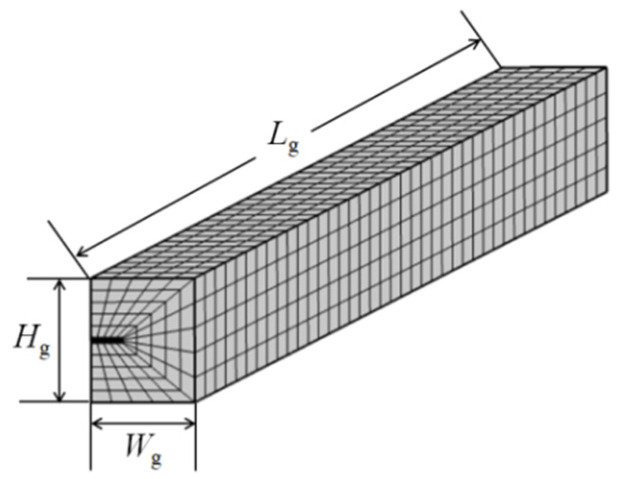
FEM model and mesh dividing of optical fiber coated by glue.

**Figure 3 sensors-22-03529-f003:**
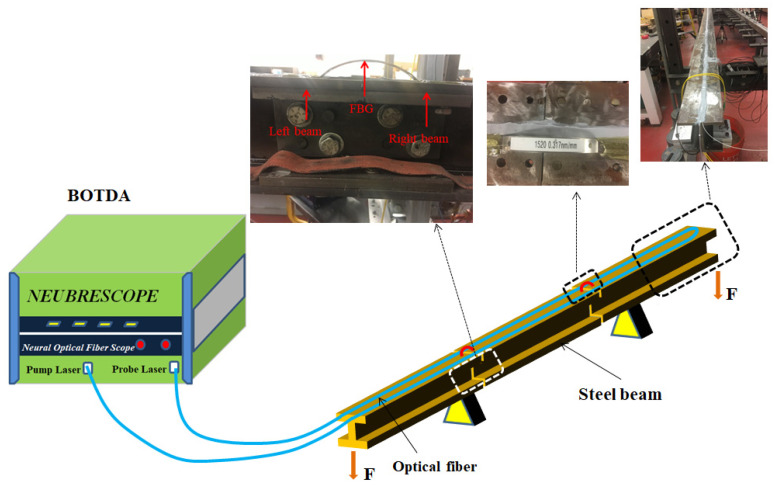
Schematic of experiment setup for detecting cracks with long distributed optical fiber sensors.

**Figure 4 sensors-22-03529-f004:**
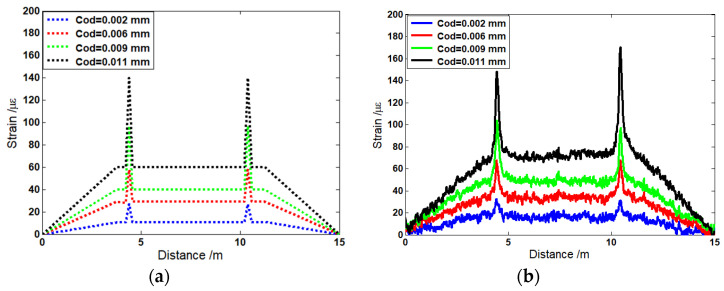
Strain curves with (**a**) theoretical and (**b**) experimental results for COD = 0.002 mm, 0.006 mm, 0.009 mm, and 0.011 mm.

**Figure 5 sensors-22-03529-f005:**
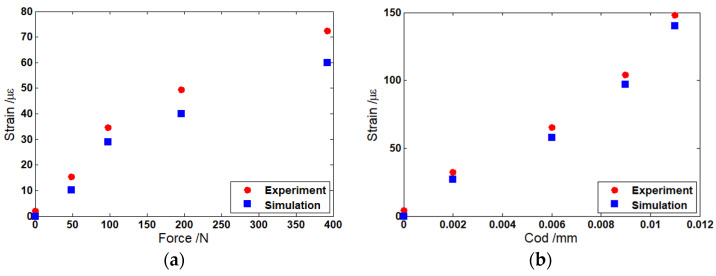
(**a**) Comparison of strain between theoretical and experimental results for *F* = 49 N, 98 N, 196 N, and 392 N. (**b**) Comparison of strain between theoretical and experimental results for COD = 0.002 mm, 0.006 mm, 0.009 mm, and 0.011 mm.

**Figure 6 sensors-22-03529-f006:**
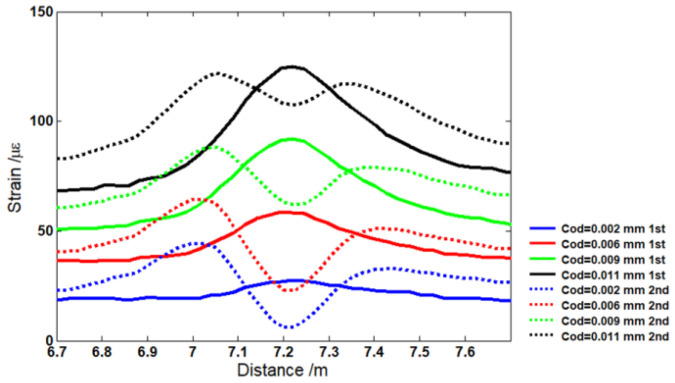
Comparison of first-time and second-time measurement strains in crack region.

**Figure 7 sensors-22-03529-f007:**
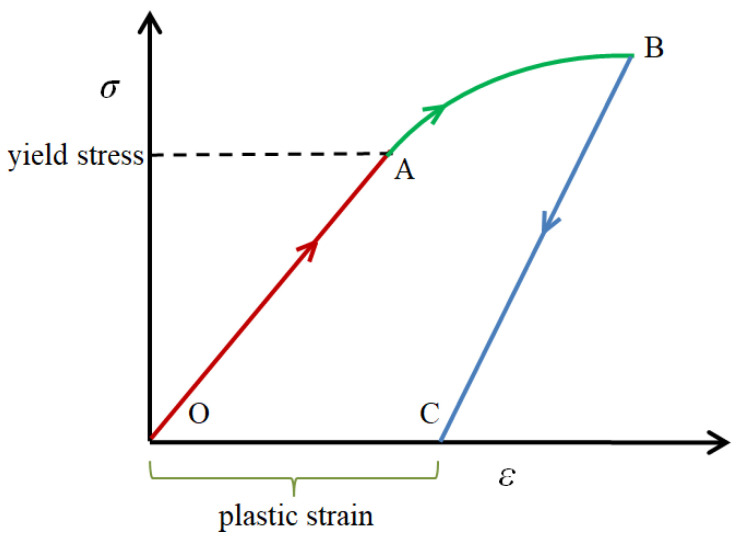
Non-linearity of distributed optical fiber.

**Table 1 sensors-22-03529-t001:** Mechanical parameters of distributed optical fiber.

Material	Young’s Modulus (MPa)	Poisson’s Ratio
Fiber core and cladding (silicon dioxide)	72,000	0.2
Fiber coating (acrylate)	4.17	0.48
Glue (epoxy resin)	4000	0.34

## Data Availability

Not applicable.

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
