# Peer review of "Theoretical and Experimental Studies of Micro-Surface Crack Detections Based on BOTDA"

_sensors, 2022, doi:10.3390/s22093529_

Round 1
Reviewer 1 Report
The paper is good but much to brief although there is no page limit: starting with the abstract it needs to be improved. The references list shall be expanded to 30-45 titles mainly contemporary research. The paragraph entitled "model analysis" shall detail materials and methods properly. For example figure 3 is insuficiently described. The section on results and discussion is best done. The conclusions again need expansion by reiterating the main findings.
Reviewer 2 Report
The subject area of this paper is long standing and there are a very large number of known results developed by a variety of methods. This paper fails to make the case as to why more are needed. Also there is little new in how the results given are actually derived. Hence there are no real new contributions to either theory or applications of the level necessary to merit learned journal publications. The paper is also too long relative to its content. The results in this paper are predictable, i.e. follow naturally from known. I cannot see any major advances and the authors clearly do not show one area where these results are superior or the only way to go.
Reviewer 3 Report
Interesting work. However, I propose the authors to perform a thorough reading to correct the simple language mistakes.
The biggest revision I would like to see is on the introduction and the results. What is the state of the art in the use of optical fibers for monitoring of crack openings? Explanation on the crack openings and the resulting strains in the introduction is not sufficient. What kind of cracks do you have in such applications? What is the problem you are trying to solve? And at the end, did you manage to solve it? The importance of the study is not elaborated.
Results part: Was the fiber material non-linearity and the deformation beyond yield problem reported in the literature for such small strains?
Other remarks are typed in the attached file.

Round 2
Reviewer 1 Report
The highlight of the original contribution on including the theoretical-experimental comparison, typical for civil engineering research, when experimental research is also possible, is appreciated.
The extent of detail is now satisfactory.
Reviewer 2 Report
The new version has been more clearly written and the results have been more rigorously proved. In my opinion, this paper can be further improved in the following aspects:
1. The contributions should be more clearly explained with more details on how to improve the existing results, especially in the references the authors cited.
2. Some future directions can be discussed in the conclusion part.
3. Some latest references about computer vision should be added to give readers an up-to-date picture. In this sense, the following papers can be referred: A small-sized object detection oriented multi-scale feature fusion approach with application to defect detection, IEEE Transactions on Instrumentation and Measurement; FMD-Yolo: An efficient face mask detection method for COVID-19 prevention and control in public, Image and Vision Computing.
4. The authors still need a careful check of English, formulas and format/style.
Reviewer 3 Report
Thank you for the edits. I think the results part can be improved by more explanation and referral to similar studies. For example, around line 164 it would be nice to add references where similar observations were done by other researchers, and explain why nonlinearity gives such effect.
